# Kidney Injuries and Evolution of Chronic Kidney Diseases Due to Neonatal Hyperoxia Exposure Based on Animal Studies

**DOI:** 10.3390/ijms23158492

**Published:** 2022-07-31

**Authors:** Liang-Ti Huang, Chung-Ming Chen

**Affiliations:** 1Department of Pediatrics, Wan Fang Hospital, Taipei Medical University, Taipei 110, Taiwan; a9309@tmu.edu.tw; 2Department of Pediatrics, School of Medicine, College of Medicine, Taipei Medical University, 250 Wu-Hsing Street, Taipei 110, Taiwan; 3Department of Pediatrics, Taipei Medical University Hospital, Taipei 110, Taiwan

**Keywords:** chronic kidney disease, hyperoxia, kidney injury, nephrogenesis, kidney fibrosis, prematurity

## Abstract

Preterm birth interrupts the development and maturation of the kidneys during the critical growth period. The kidneys can also exhibit structural defects and functional impairment due to hyperoxia, as demonstrated by various animal studies. Furthermore, hyperoxia during nephrogenesis impairs renal tubular development and induces glomerular and tubular injuries, which manifest as renal corpuscle enlargement, renal tubular necrosis, interstitial inflammation, and kidney fibrosis. Preterm birth along with hyperoxia exposure induces a pathological predisposition to chronic kidney disease. Hyperoxia-induced kidney injuries are influenced by several molecular factors, including hypoxia-inducible factor-1α and interleukin-6/Smad2/transforming growth factor-β, and Wnt/β-catenin signaling pathways; these are key to cell proliferation, tissue inflammation, and cell membrane repair. Hyperoxia-induced oxidative stress is characterized by the attenuation or the induction of multiple molecular factors associated with kidney damage. This review focuses on the molecular pathways involved in the pathogenesis of hyperoxia-induced kidney injuries to establish a framework for potential interventions.

## 1. Introduction

The prevalence rate of preterm birth is approximately 15 million infants worldwide each year. A recent cohort study reported that the incidence rate of chronic kidney disease (CKD) by gestational age at birth is 9.24 per 1,000,000 for preterm infants (<28 weeks), indicating a threefold risk of CKD with respect to term infants; no difference was found between male and female infants [1]. Preterm births vary in terms of gestational age and causes, with inadequate gestational age being a common cause. Furthermore, an adverse intrauterine environment causes 38% of preterm births, and other conditions such as preeclampsia, multiple births, and chorioamnionitis may also be involved [2]. Exposure postnatally to factors such as high oxygen concentrations [3], medications [4], and inadequate nutrition [5] likely adversely influence postnatal growth and ongoing organ development. The renal consequences of preterm births have attracted increasing attention and include a high risk of (CKD) [1], a quick progression of renal pathology [6], and predisposition toward hypertension [7]. The third trimester of pregnancy is the most active period of fetal nephrogenesis, during which more than 60% of nephrons are formed [8,9]. Preterm birth (within <37 gestational weeks) interrupts the development and maturation of the kidneys during the critical growth period. Neonates born preterm have an immature antioxidant defense system [10] and present an imbalance between the oxidant and the antioxidant system leading to an increased level of free radicals (FR), with subsequent oxidative damage to organs [11]. Moreover, oxygen resuscitation [12] and intensive care maneuvers such as assisted ventilation, surfactant administration [13], total parenteral nutrition [14], and blood transfusions [15] enhance FR production, which further increments oxidative stress. Experimental hyperoxic exposure has been used in various previous studies to investigate the effects of oxidative stress on preterm neonates [16,17]. Hyperoxia during the neonatal period impairs renal tubular development. Human and animal studies have demonstrated that neonatal hyperoxia increases oxidative stress and induces glomerular and tubular injuries, which are manifested as renal corpuscle enlargement, renal tubular necrosis, interstitial inflammation, and kidney fibrosis during the perinatal period [18,19,20,21,22]. Through the suppression of antioxidants such as glutathione peroxidase, catalase, and superoxide dismutase activity [23,24], hyperoxia exposure augments free radical and reactive oxygen species (ROS) production, which leads to increased oxidative stress. Human kidney development is completed in utero in 36 gestational weeks [25], whereas nephrogenesis in rats begins on the 12th embryonic day and is completed within 10 to 15 days after birth [26]. Rats are born with immature kidneys; the first two postnatal weeks correspond to the second and third trimesters of human pregnancy during which fetal nephrogenesis occurs. Studies conducted using neonatal rat models have demonstrated the mechanism of fetal nephrogenesis in humans. Although previous studies have clearly demonstrated the presence of renal anomalies after hyperoxic exposure, the corresponding renal diseases and progression to CKD have yet to be investigated. Thus, this review focuses on CKD among prematurely born infants who were exposed to hyperoxia and discusses the theory, experimental evidence, and indicators reported in the literature. In general, vigilant surveillance and therapeutic interventions must be available to these infants.

## 2. Experimental Oxygen Studies and Kidney Injury

Hyperoxia causes structural defects and functional impairment [22,27,28,29,30,31,32], thereby predisposing newborns to hypertension and CKD [33,34,35]. Studies have demonstrated—using different animal models, including rat and mice models—that exposure to hyperoxia can cause renal corpuscle enlargement, tubular injuries, reduced glomerular filtration rate (GFR), and electrolyte derailment, as well as increased collagen content, kidney fibrosis, and apoptosis [21,22,36] (Table 1). In a mouse model based on hyperoxia gas exposure (85% O_2_) in the early neonatal period (postnatal days 1 to 28), Mohr et al. showed that hyperoxia-induced neonatal renal injuries involve two phases: the acute and the regenerative phases. The acute renal injury phase is characterized by deregulated mitochondrial biogenesis, reduced renal cortical growth, reduced cell proliferation, and the activation of interleukin-6/Stat3 signaling coupled with active Smad2/transforming growth factor-β (TGF-β) signaling. The regenerative phase is characterized by renal cortical catch-up growth with reduced glomeruli relative to the renal cortical area, thereby causing glomerular and tubular dysfunction [37]. A previously developed rat model showed that hyperoxia exposure (80% O_2_) in the early neonatal period (postnatal days 3–10) was associated with a 25% reduction in nephron number in adulthood [38]. Popeseu et al. also discovered a considerable decrease in both nephrogenic zone width and glomerular diameter and a considerable increase in apoptotic cell count using a similar rat model [36]. Compared with rats exposed to ambient air, rats exposed to neonatal hyperoxia (85% O_2_ for 14 days) exhibited a notably lower glomerular number, a higher kidney injury score, and elevated expression of toll-like receptor 4 (TLR-4), myeloperoxidase (MPO), and 8-hydroxy-2-deoxyguanosine (8-OHdG) [39]. A similar study demonstrated that newborn rat exposure to hyperoxia caused tubular atrophy, dilatation of the tubular lumen, vacuolar degeneration of the tubular epithelium, and increased space between the renal tubules in conjunction with collagen deposition and macrophage infiltration [40]. In both short-term (1 week in Chen and in Chou et al.) [39,40] and long-term (3 weeks in Jiang et al.; 95% O_2_ for 7 days and 60% O_2_ for following 14 days) [22] studies, hyperoxic exposure resulted in increased collagen I and tissue connective tissue growth factor (CTGF) expression leading to renal tubular damage. In addition, the results of long-term studies on the effects of neonatal rats exposed to hyperoxia (85% O_2_ for 14 days, then 21% O_2_ till P 60 d) indicated impaired proximal tubular development [41,42], increased glomerular injury [43], decreased glomerular volume, and decreased nephron number [44]. Thus, neonatal hyperoxia exposure may cause transient apoptosis, renal fibrosis, and long-term glomerular and tubular damage and dysfunction in the kidneys. In contrast, however, in a mouse model, exposure to 65% O_2_ from postnatal day 1 to 7, and to >98% O_2_ from day 0 to 4 then to 21% for the following 4 days, did not lead to any significant alterations in renal development or nephron number [18,21]. The contribution of hyperoxia-induced neonatal kidney damage to vulnerability to adult CKD has yet to be clearly understood.

## 3. Predisposition to CKD Due to Hyperoxia-Induced Kidney Injuries

### 3.1. Proximal Tubular Injury and Interstitial Fibrosis

CKD affecting 13% of the global population is a growing epidemic resulting from renal insults caused by, e.g., hypertension, inflammatory glomerular diseases, diabetes mellitus, genetic disorders, and toxins [45]. Although CKD can be triggered by various factors, glomerulosclerosis, vascular sclerosis, and tubulointerstitial fibrosis are common in CKD, indicating that progressive injury occurs through a common final pathway [46]. CKD is the common endpoint of different etiologies including glomerular injuries, repeated acute kidney injury (AKI), and chronic tubulointerstitial injures. The progression of CKD follows a common pathway whereby the normal renal parenchyma is replaced with matrix proteins such as collagen I, III, IV and fibronectin. The basic functional unit of the kidney is the nephron which filters the blood in the glomerulus. The resulting ultrafiltrate passes through the tubules, composed of a highly specialized epithelium, which reabsorb water and electrolytes [47]. With regard to histopathology, tubulointerstitial fibrosis (TIF) is characterized by extracellular matrix (ECM) accumulation, tubular atrophy, inflammatory cell infiltration, and peritubular microvascular loss, which are typically observed in CKD [48,49]. Matrix proteins accumulation in the glomerulus is termed glomerulosclerosis, whereas TIF describes the presence of matrix proteins replacing the tubules and/or surrounding the interstitium. The tubular epithelium, especially in the proximal tubules, is affected by acute and chronic injuries. The injured epithelium dedifferentiates and proliferates, thereby promoting healing after AKI [50]. Repetitive or persistent epithelial injuries may cause tubular apoptosis, which leads to progressive TIF. The dying epithelial cells may produce proinflammatory cytokines and other growth factors that promote inflammation and fibrosis. Renal fibrosis is typically preceded by the infiltration of inflammatory cells, including lymphocytes, monocytes/macrophages, dendritic cells, and mast cells. Fibrosis is primarily promoted by persistent inflammation [51]. Experimental models and human pathological studies of TIF have demonstrated that the activation of tubular epithelial cells and interstitial fibroblasts causes the excessive generation of ECM, which is constituted mostly by collagen, and the formation of scar tissue. Injured proximal tubular epithelial cells synthesize collagen, which promotes basement membrane thickening (collagen IV) and interstitial fibrosis (collagen I, III, IV) [52]. The dedifferentiated epithelial cells exhibit a partially mesenchymal phenotype, which is associated with an increased production of profibrotic cytokines. Renal injury alters the epithelial cell cycle, and these cell cycle changes may also impact TIF progression and the transition from AKI to CKD [47]. Most proximal tubule epithelial cells in uninjured kidney are quiescent (cell cycle stage G0). However, during injury, the cells may enter the cell cycle (G1, S, G2, M) to help replace cells lost to apoptosis/necrosis, and some cells become arrested in either G1 or G2. Cell arrest is adaptive to allow time for repair of any DNA damage and prevent the propagation of mutations that occur in injured cells. Chronically injured epithelial cells are arrested in G2/M; this cell cycle dysfunction is also associated with the excessive production of profibrotic growth factors [53]. G2/M cell cycle arrest results in increased activity of c-jun N-terminal kinase (JNK), which promotes the production of TGF-β and CTGF/CCN2 [53]. The augmented production of epithelial TGF-β may induce the interaction of other growth factors with fibroblasts, thereby promoting TIF progression.

### 3.2. Nephron Number Loss and Increase in Glomeruli Diameter

Studies have revealed that early exposure to hyperoxia causes a decrease in nephron number and an increase in glomeruli diameter as well as an increase in the incidence of glomerular injuries and in the number of apoptotic cells [37,38,43]. As nephron number decreases, the filtration capacity of nephrons increases. The number of functional nephrons decreases, creating more metabolic work for the remaining tubules. In agreement with Brenner’s hypothesis about declining renal function and glomerular hyperfiltration [54], the remaining proximal tubules become hypertrophic to meet the increased demand of water and solutes reabsorption [55]. Although total energy expenditure in the CKD kidney decreases, the metabolism of the surviving nephrons increases to support the compensatory changes in reabsorption [56,57]. Protein intake correlates positively with proximal tubule hypertrophy and, in the rat subtotal nephrectomy model, led to more oxidative stress [58]. The increase in perfusion pressure above the normal physiological range may affect the structural filtration stability and cause glomerulosclerosis and nephron loss [59]. In a study on the effects of a low-protein diet during pregnancy and lactation, reduced nephron number, increased glomerulus volume, and enhanced TGF-β and collagen I expression and STAT3 phosphorylation were observed in children [60]. Srivastava et al. investigated the role of biomechanical forces in hyperfiltration-mediated glomerular injuries due to congenital anomalies in the kidney and urinary tract (CAKUT) by analyzing the effects of biomechanical forces on glomerular podocytes [61]. Moreover, children with CAKUT may progressively develop CKD because of early renal changes due to maladaptive hyperfiltration that leads to increased fluid flow shear stress [61]. Although the long-term consequences of preterm birth in the kidneys have yet to be discovered, evolutionary adaption may explain other responses in the CKD kidney that are initially adaptive but lead to dysfunction, such as senescence and inflammation. The decrease in nephron number and proliferation of tubular cells can induce functional changes in the kidneys later in life [37]. The available evidence indicates that preterm birth increases the risk of CKD [30,32], and neonatal hyperoxia exposure induces changes in renal structure and a decrease in GFR as well as glomerular and tubular damage, thereby increasing the chances of renal disorders in adulthood.

### 3.3. Glomerular and Podocyte Injury

The glomerulus contains four different cell types. Fenestrated endothelial cells cover the glomerular basement membrane (GBM) from the vascular side, and podocytes from the outside (urinary side), all together building the glomerular filtration barrier [62]. Mesangial cells produce extracellular matrix and scaffold the capillaries. The fourth cell type, parietal epithelial cells, lines the outer aspect of the glomerulus and thereby defines the Bowman’s space from where primary urine flows into the tubule [63]. Mature podocytes do not proliferate in vivo due to their highly differentiated phenotype; thus, they respond to different types of injury through detachment from the glomerular basement membrane, dedifferentiation, autophagy, and apoptosis [64]. The mechanisms underlying podocyte injury are complex and include hemodynamic and metabolic pathways as well as the interplay of vasoactive molecules, growth factors, and cytokines [64,65]. Accumulating experimental and clinical evidence suggests that podocytes are quite vulnerable to oxidative damage, and amplification of oxidative stress seems to be a final and common pathway shared by different aggressors at the cellular level. Glomerular injury leads to podocyte loss and induces adhesion between the glomerular basement membrane and the parietal epithelium. A protein-rich ultrafiltrate usually passes from the capillaries to the Bowmans space and tubular lumen, but increased adhesion in the tuft and breaks in the parietal epithelium misdirect the protein ultrafiltrate to the interstitium [66]. Podocyte injury can induce their detachment from the GBM and thus their loss, which is the major determinant of progressive glomerular injury and glomerulosclerosis [67]. The loss of podocytes will also induce endothelial stress and injury, since podocyte-derived angiogenic factors such as VEGF are essential for normal endothelial homeostasis [63]. Paracrine signaling from injured podocytes induces the activation of parietal epithelial cells and mesangial cells, which directly facilitates the onset of glomerulosclerosis. Glomerular injury increases the filtered load of albumin, and this augmented proximal tubular protein reabsorption can have both direct toxic and paracrine effects that may promote TIF, which leads to CKD. 

### 3.4. Similarities and Differences between Hyperoxia- and Other Factors-Induced Pathomechanisms in CKD

There are some similarities and differences in CKD induced by hyperoxia or by other factors, such as diabetes, hypertension, or obstructive uropapthy. CKD induced by hyperoxia and other factors CKD is characterized by nephron loss, nephron hypertrophy, proximal tubular injury, and intersitial fibrosis, which are originated from oxidative stress and inflammatory processes. CKD due to other factors presents dominant podocyte injury and detachment, focal segmental glomerulosclerosis, global glomerulosclerosis, and subsequent nephron atrophy, which are caused by direct toxicity, local or systemic infection, and genetic factors. Nephron loss involves a nonspecific wound-healing response that includes interstitial fibrosis. Infitrating immune cells, albuminuria, and, in diabetes, glucosuria, activate proximal tubular epithelial cells, resulting in the secretion of proinflammatory and profibrotic mediators, and promote interstitial inflammation and fibrosis [68]. The increased tubular transport load of remnant nephrons also involves anerobic metabolism, intracelluar acidosis, and endoplasmic reticulum stress, which promote secondary tubular injury [69]. In other forms of CKD, angiotensin II plays an important role through the mechanistic targeting of rapamycin signaling, maintaining persistent podocyte hypertrophy and glomerular hyperfiltration and ultimately aggravating podocyte loss and proteinuria. Angiotensin II possibly also contributes to the dysregulated response of progenitor parietal epithelial cells along Bowman’s capsule, generating FSGS lesions instead of replacing lost podocytes [70]. This structural remodelling of the glomerulus presents clinically as proteinuria, which is a marker of nephron damage and is predictive of CKD progression [71].

## 4. Cellular and Molecular Aspects

### 4.1. Influence of Hyperoxia-Inducible Factor-1a (HIF-1α) on Tubular Development

Several biomolecular factors including TGF-β, HIF-1α, and integrin-linked kinase have been found to influence in vivo epithelial–mesenchymal transformation [72,73,74]. The prevention of TGF-β-mediated signaling by bone morphogenetic protein-7 aids the reversal of TGF-β-mediated TIF [72] (Figure 1).

Hyperoxia-inducible factors (HIFs) are heterodimers comprising one of three O_2_-sensitive alpha subunits (HIF-1α, HIF-2α, and HIF-3α) and a constitutively expressed beta-subunit HIF-1β [75]. HIF-1α and HIF-1β are the key regulators of oxygen homeostasis. HIF-1α promotes organogenesis [76,77] by regulating the expression of various factors involved in angiogenesis, cellular proliferation, and apoptosis [78,79]. Within the developing kidney, HIF-1α is expressed weakly in the outer cortex and strongly in some tubular and collecting duct epithelial cells. Mice genetically deficient in HIF-1α die at midgestation (embryonic day 9.5 [E9.5]) show vascular and neural tube defects [76,80]. Popeseu et al. investigated the influence of transient hyperoxia exposure on nephrogenesis in neonatal rats. Transient hyperoxia during nephrogenesis suppressed HIF-1α, reduced nephrogenic zone width and glomerular diameter, and increased apoptotic cell count, which were reversed by a HIF-1α stabilizer; however, no changes were observed in nephron number after nephrogenesis [36]. Xu et al. discovered that neonatal hyperoxia impaired proximal tubular development in conjunction with downregulated HIF-1α and altered mitogen-activated protein kinase (MAPK)/extracellular signal-regulated kinase (ERK) signaling and catalase [41]. MAPK/ERK signaling is crucial to the proliferation and differentiation of nephron progenitors. The catalase enzyme is key to protecting the cell from oxidative damage due to ROS [81]. Nephrectomized animals exhibited significant elevated levels of proinflammatory and profibrotic factors, activation of NF-κB, and increased activation of TGF-β/Smad3 and MAPK signaling pathways [82]. ROS activate a broad variety of hypertrophy signaling kinases and transcription factors, such as MAPK and NF-κB, as well as proliferation of activation of matrix metalloproteinases (MMPs) [83]. Thus, early exposure to hyperoxia can impair nephrogenesis and induce proximal tubular injuries through HIF-1α attenuation, predisposing infants to adult CKD.

### 4.2. Role of the Proinflammatory Cytokine Interleukin-6

Cytokines aid the inflammatory response but may induce organ dysfunction when released in excess [84,85]. IL-6 is a pleiotropic cytokine that not only regulates immune and inflammatory responses but also affects hematopoiesis, metabolism, and organ development [86]. The level of IL-6 is high in the renal tissue of patients with kidney diseases, diabetic nephropathy, glomerulonephritis, or obstructive nephropathy. Renal cells that express and secrete IL-6 include podocytes, endothelial cells, mesangial cells, and tubular epithelial cells. IL-6 signaling can enhance cell proliferation, affect differentiation, and promote TIF [86]. After prolonged exposure to oxygen, the levels of IL-6, TGF-β, and 8-OHdG increase in the tracheal aspirates of infants, leading to chronic lung disease [87,88,89]. Neonatal rat exposed to hyperoxia also augmented the production of 8-OHdG and TLR-4, MPO activity, and the levels of IL-1β, NF-κB, M1 and M2 macrophage and collagen in the kidneys, which indicated increased oxidative stress (8-OHdG), neutrophil accumulation (MPO activity), inflammatory response (TLR-4, IL-1b, NF-kB, and M1 and M2 macrophage), and fibrotic reaction (collagen) [39,40]. Furthermore, neonatal hyperoxia exposure considerably increased IL-6 levels in the lungs [90,91], indicating that hyperoxia enhanced ROS production, and the subsequent oxidative stress induced an inflammatory cellular response [92]. Mohr et al. discovered that elevated levels of IL-6 induced transient changes in renal fibrosis and a considerable decrease in GFR, glomerular number, and cell proliferation in neonatal mice after prolonged exposure to oxygen. These changes did not occur in IL-6^−/−^ animals [37]. Increased renal expression of IL-6 and a marked activation of IL-6/Stat3 signaling in the acute renal injury phase may thus remain unchanged after regeneration under normoxic conditions [37]. In addition to increasing IL-6 levels, hyperoxia induces a marked activation of Smad2 and subsequently increases the expression of plasminogen activator inhibitor-1 (PAI-1) and other downstream molecules, such as CTGF, collagen a-4 type IV (collagen IV), and elastin. Smad2, an indicator of the TGF-β pathway activation, enhances the synthesis of matrix proteins and the secretion of protease inhibitors, which results in matrix accumulation. TGF-β is a key profibrotic growth factor that is activated in AKI and is associated with cellular responses that lead to the development of CKD. TGF-β signaling may sustain proximal tubule injury by inducing cell dedifferentiation, cell cycle arrest, and increased susceptibility to apoptosis. Furthermore, TGF-β signaling promotes macrophage chemotaxis, endothelial injury, and myofibroblast differentiation after AKI [93]. Tubular overexpression of TGF-β led to tubular autophagy and degeneration as well as peritubular fibrosis [94]. PAI-1, CTGF, collagen IV, and smooth muscle alpha-actin are downstream profibrotic effectors of Smad2 and are associated with enhanced extracellular matrix remodeling and fibrosis; they are also associated with damage in several organs in humans [95,96,97,98,99]. The activation of both IL-6 and TGF-β signaling due to hyperoxia during a critical period of nephrogenesis could adversely affect energy metabolism, tubular growth, and nephron functioning, thereby predisposing infants to adult CKD.

### 4.3. Influence of Hyperoxia on Nephrogenesis and Renal Fibrosis through Wnt/β-Catenin Signaling

Wnts are a family of secretory proteins that, upon binding with their cellular membrane receptors, induce a series of downstream signaling events resulting in the phosphorylation of beta-catenin [100]. The Wnt/β-catenin signaling pathway is crucial for organ development, tissue homeostasis, and postinjury organ repair [101,102,103]. Transient Wnt/β-catenin signaling stimulates repair and regeneration after AKI, and the continuous activation of the pathway promotes CKD-related fibrogenesis [104]. The Wnt/β-catenin signaling pathway is inactive in normal kidneys but is activated after kidney injury in various CKD situations, such as in the presence of ischemia–reperfusion injury, 5/6 nephrectomy, unilateral ureteral obstruction, and doxorubicin nephropathy [105,106,107]. The upregulation of multiple Wnt ligands and the activation of β-catenin are observed in renal biopsies of individuals with diabetic nephropathy, IgA nephropathy, and lupus nephritis and make them prone to podocyte damage and oxidative stress [108,109,110]. Previous evidence suggests that the action of Wnt/β-catenin is dependent on the activation of receptor of advanced glycation end product (RAGE) by advanced oxidation protein products (AOPPs). This triggers a cascade of reactions, including induction of NADPH oxidase, generation of ROS, and NF-κB activation, which leads to the induction of Wnt ligands and the activation of β-catenin [111]. Zhou et al. discovered that Wnt/β-catenin is activated in CKD patients and mediates oxidative stress-triggered podocyte dysfunction and proteinuria through a series of signaling events involving the induction of ROS/p65 NF-κB and Wnt [100]. Lin et al. experimentally demonstrated that the suppressed antioxidant element Se may induce renal fibrosis by enhancing the Wnt/β-catenin signaling pathway [112]. Thus, oxidative stress may lead to CKD and kidney fibrosis through the activation of the Wnt/β-catenin signaling pathway.

## 5. Therapeutic Approach According to Molecular Markers

In addition to basic science studies elucidating the importance of kidney injury in hyperoxia-exposed animals, other studies have described a new class of therapeutic interventions or medications that target molecular factors and protect against hyperoxia-induced kidney injury in neonatal animals. Experimentally systemic administration of the HIF-1α stabilizer dimethyloxalylglycine (DMOG) resulted in enhanced expression of HIF-1α and improved nephrogenesis: kidneys from hyperoxia-exposed pups treated with DMOG exhibited a nephrogenic zone width and a glomerular diameter similar to those of controls [36]. Investigations also showed that cathelicidin treatment attenuated kidney injury, as evidenced by lower kidney injury scores, less 8-OHdG-positive cells, reduced collagen deposition, and reversion of hyperoxia-induced M1/M2 macrophage polarization, accompanied by decreased NF-kB levels [40]. IL-6 deficiency ameliorated experimental hyperoxia-induced glomerular and tubular dysfunction through the inhibition of inflammatory signaling. These observations may indicate new avenues to protect premature infants from CKD [37]. In a rat model of hyperoxia-induced lung injury, resveratrol alleviated hyperoxia-induced histological injury in the lungs, regulated the redox balance, decreased proinflammatory cytokine release, and downregulated the expression of fibrosis-associated proteins through Wnt/β-catenin signaling suppression [23]. These novel therapeutic approaches provided some targeting goals to reduce the risk of CKD in premature babies. Nevertheless, there are currently no effective approaches for neonates exposed to hyperoxia. Large studies in specifically selected patient cohorts with current endpoints will require a multicenter approach, with long recruitment time and duration, which will substantially increase the time and costs.

## 6. Future Studies on Hyperoxia-Induced Kidney Injuries

Glomerular and tubular injury and renal fibrosis are not a single process but indicate various pathologic and fibrotic processes in specific renal compartments with a particular cellular and molecular composition. They involve virtually all renal cells and a large number of molecules, albeit only few molecules are directly involved in fibrogenesis itself. Hyperoxia during the neonatal period in animals causes pathological changes including glomerular injuries and proximal tubular dysfunction. Kidney injuries due to hyperoxia exposure vary, depending on the injury phase (acute or regenerative) after recovery in ambient air. The results of long-term studies indicate that early exposure to hyperoxia results in glomerular reduction and proximal tubular dysfunction [28], whereas short-term studies have not demonstrated the occurrence of deleterious effects [10]. In addition, several groups have shown that β-catenin activation in fibroblasts or pericytes promote TIF progression [113,114]. In contrast, investigations also demonstrated that β-catenin activity in the proximal tubule protect against TIF [115]. Although an increment in HIF-1α expression improved nephrogenesis and hyperoxia-induced kidney injury [38], increased HIF expression was found in animal models of CKD and in renal biopsy material from patients with diabetic nephropathy and other forms of renal disease [116,117]. Another study identified epithelial HIF-1 as a promoter of renal fibrosis in experimental unilateral ureteral obstruction [73]. Thus, the implications of these results are unclear because of a lack of long-term cohort studies specifying the target markers. Although pathological changes in glomeruli and tubules and the enhancement of biomolecular pathways may be the major causes of CKD, further studies must investigate the correlation between damaged nephrons and renal interstitial fibrosis. Furthermore, advanced therapeutic strategies must be developed to prevent CKD in prematurely born infants.

In conclusion, preterm birth and low birth weight contribute to the increased risk of CKD in later life. Oxygen therapy to treat neonatal respiratory diseases can also lead to kidney injury during nephrogenesis. The injured kidney structure and dysfunction may lead to CKD through ROS- and inflammation-induced nephron loss, proximal tubular and glomerular injuries, and interstitial fibrosis. These mechanisms are similar to those activated by other factors that may cause CKD. Several molecular signaling pathways are involved in the pathogenesis of hyperoxia-induced CKD. Targeting pathomolecular regulators might provide the means to simultaneously regulate the activity of multiple pathways that coordinately regulate different aspects of CKD progression.

## Figures and Tables

**Figure 1 ijms-23-08492-f001:**
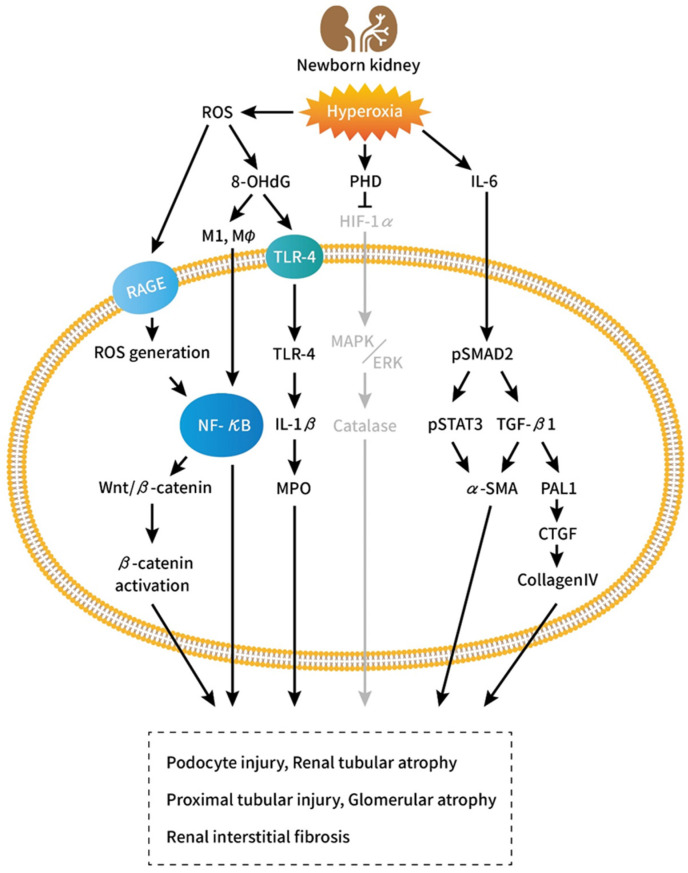
Schematic of the mechanism of hyperoxia-induced kidney injury and renal interstitial fibrosis. Hyperoxia induces reactive oxidative stress; prolyl hydroxylases domain enzymes, which downregulate HIF-1α, upregulate IL-6, and trigger the activation of 8-OHdG, bind to the plasma membrane receptor RAGE and TLR-4, thereby enhancing the generation of ROS. This regulates the inflammatory transcription factors NF-κB, IL-1β, and MPO, which induce Wnt ligands and subsequently activate β-catenin. The activation of β-catenin triggers podocyte injury through dedifferentiation and mesenchymal transition. The downregulation of HIF-1α suppresses MAPK and ERK, thereby reducing catalase levels. The reduction of catalase triggers proximal tubular injury and impairs glomerular development. The upregulation of IL-6 triggers the activation of phosphorylated Smad2. This in turn induces the phosphorylation of STAT3, the activation of TGF-β1, and the formation of alpha SMA, PAL1, CTGF, and collagen IV. The activation of α-SMA and collagen IV subsequently induces renal mesangial and interstitial fibrosis. Abbreviations: PHD: prolyl hydroxylases domain enzyme; IL-6: interleukin-6; PAI-1: plasminogen activator inhibitor-1; CTGF: connective tissue growth factor; HIF-1α: hypoxia inducible factor-1α; 8-OHdG: 8-hydroxydeoxyguanosine; MPO: myeloperoxidase; TLR4: toll like receptor; IL-1β: interleukin-1β; MAPK/ERK: mitogen-activated protein kinase, extracellular signal-regulated kinase; TNF-α: tumor necrosis factor-α; STAT3: signal transducer and activator of transcription3; TGF-β1: transforming growth factor-β1; alpha SMA: alpha smooth muscle actin.

**Table 1 ijms-23-08492-t001:** Experimental models of hyperoxia exposure and kidney injury.

Model	Species	Primary Target Lesion	Molecules	Ref
85% O_2_, P1 to P28, 21% O_2_ till P70 (Mohr et al.)	mouse	Glomerular filtration rate Kidney cortex area Glomerular number Glomerular diameter Proximal tubular proliferation	IL-6 Collagen IV PAI-1 CTGF Smad2	[37]
80% O_2_ P3 to P10 (Popeseu et al.)	rat	Nephrogenic zone Glomerular diameter Glomerular apoptotic cells	HIF-1α	[36]
95% O_2_, P1 to P7, 60% O_2_ till P21 (Jiang et al.)	rat	Tubular injury score Glomerular size	Total collagen 8-oHdG CTGF	[22]
80% O_2_, P1 to P14 (Chen et al.)	rat	Kidney injury score Glomerular number Glomerular injury score	8-OHdG MPO activity TLR4 IL-1β	[39]
85% O_2_, P1 to P7 (Chou et al.)	rat	Tubular injury score	M1 macrophage 8-OHdG Collagen NF-κB	[40]
85% O_2_, P0 to P14; 21% O_2_ till P60 (Xu et al.)	rat	Nephrogenic zone Epithelial cells of mature proximal tubules Tubular cell apoptosis	MAPK/ERK HIF-1α Catalase IL-6 TNF-α Claudin-4 Occludin Zonula occluden-1 (ZO-1)	[41,42]
65% O_2_, P1 to P7; 21% O_2_ till P56 and P10m (Sutherland et al.)	mouse	Nephron number Renal corpuscles	-	[21]
80% O_2_, P3 to P10; 21% O_2_ till P11ms (Sutherland et al.)	rat	Glomerular injury Creatinine clearance	-	[43]
85% O_2_, P3 to P15; 21% O_2_ till P9ms (Kumar et al.)	mouse	Glomerular diameter Glomerular volume Nephron number	-	[44]
80% O_2_, P3 to P10; 21% O_2_ till P15wks (Yzydorczyk et al.)	rat	Blood pressure Microvascular rarefaction Nephron number	Superoxide dismutase analogue	[38]
>98% O_2_ P0 to P4; 21% O_2_ till P5, P8 (Torbati et al.)	rat	Tubular necrosis, dilation, and degeneration, Interstitial inflammation	-	[18]

Abbreviations: IL-6: interleukin-6; PAI-1: plasminogen activator inhibitor-1; CTGF: connective tissue growth factor; HIF-1α: hypoxia inducible factor-1α; 8-OHdG: 8-hydroxydeoxyguanosine; MPO: myeloperoxidase; TLR4: toll like receptor; IL-1β: interleukin-1β; MAPK/ERK: mitogen-activated protein kinase, extracellular signal-regulated kinase; TNF-α: tumor necrosis factor-α.

## Data Availability

Data sharing is not applicable to this article.

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
