# Peer review of "Kidney Injuries and Evolution of Chronic Kidney Diseases Due to Neonatal Hyperoxia Exposure Based on Animal Studies"

_ijms, 2022, doi:10.3390/ijms23158492_

Round 1

Reviewer 1 Report

Well written article. Should be a valuable addition to the literature. 

Author Response

Response to Reviewer 1 Comments

Point 1: Well written article. Should be a valuable addition to the literature.

Response 1: Thanks for reviewer’s comments.

Reviewer 2 Report

Huang et al. summarize the pathomechanisms and molecular pathways of CKD induced by hyperoxia in neonates based on animal studies. The topic is interesting and well-written, however, several information is missing in the current MS. My remarks are the following:

1. Please add epidemiologic information to the Introduction part: i) how often occur preterm birth in the general population, ii) how often develop premature newborns CKD, and iii) is there any difference in the development of CKD between males and females (if data are available).

2. Please summarize the mechanisms how hyperoxia could lead to increased oxidative stress in a section before the Experimental oxygen studies part.

3. Table 1: please put the references in a separate column at the end of the table (i.e., the last column is for the references).

4. Section 3: what is the difference between the mechanisms of hyperoxia-induced CKD or other forms of CKD? In my understanding, this section is the description of oxidative stress (and chronic inflammation)-induced pathologies in the kidneys. Please try to emphasize the similarities and differences between hyperoxia and other factors-induced pathomechanisms in CKD. It would be helpful to see a Figure about the mentioned main mechanisms leading to CKD.

5. Figure 1: please explain with full names of all abbreviations in the figure legend. Additionally, several abbreviations are missing in the figure legend (e.g., PHD). 

Author Response

Response to Reviewer 2 Comments

Huang et al. summarize the pathomechanisms and molecular pathways of CKD induced by hyperoxia in neonates based on animal studies. The topic is interesting and well-written, however, several information is missing in the current MS. My remarks are the following:

Point 1: Please add epidemiologic information to the Introduction part: i) how often occur preterm birth in the general population, ii) how often develop premature newborns CKD, and iii) is there any difference in the development of CKD between males and females (if data are available).

Response 1:. Thanks for reviewer’s comments. We have added “The prevalence rate of preterm birth is approximately 15 million infants world-wide each year. Recent cohort study demonstrated the incidence rate of chronic kidney disease (CKD) by gestational age at birth was 9.24 per 1000,000 for preterm infants (<28 weeks), which was threefold risks of CKD compared with term infants and no difference was found between male and female infants [1]” on page 1, lines 27-31.

Point 2: Please summarize the mechanisms how hyperoxia could lead to increased oxidative stress in a section before the Experimental oxygen studies part.

Response 2: We have added “Through suppression of antioxidant such as glutathione peroxidase, catalase and su-peroxide dismutase activity [23, 24], hyperoxia exposure augments free radical and reactive oxygen species (ROS) production which lead to increased oxidative stress” on page 2, lines 53-56.

Point 3: Table 1: please put the references in a separate column at the end of the table (i.e., the last column is for the references).

Response 3:.Thanks for reviewer’s comments. We have put the references in a separate column at the end of the table 1.

Point 4: Section 3: what is the difference between the mechanisms of hyperoxia-induced CKD or other forms of CKD? In my understanding, this section is the description of oxidative stress (and chronic inflammation)-induced pathologies in the kidneys. Please try to emphasize the similarities and differences between hyperoxia and other factors-induced pathomechanisms in CKD. It would be helpful to see a Figure about the mentioned main mechanisms leading to CKD.

Response 4: Thanks for reviewer’s comments. We have added “There are some similarities and differences between hyperoxia-induced and other factors, such as diabetes, hypertension, or obstructive uropapthy. Both hyperoxia and other factors-induced CKD have nephron loss, nephron hypertrophy, proximal tubular injury, and intersitial fibrosis which are originated from oxidative stress and inflammatory processes. CKD due to other factors has dominant podocyte injury and detachment, focal segmental glomerulosclerosis, global glomerulosclerosis, and subsequent nephron atrophy which are sourced from direct toxicity, local or systemic infection, and genetic factors. Nephron loss involves nonspecific wound-healing response that include interstitial fibrosis. Infitrating immune cells, albuminuria, and, in diabetes, glucosuria, active proximal tubular epithelial cells, resulting in the secretion of proinflammatory and profibrotic mediators and promote interstitial inflammation and fibrosis [72]. The increased tubular transport load of remnant nephrons also involves anerobic metabolism, intracelluar acidosis, and endoplasmic reticulum stress, which promote secondary tubular injury [73]. In other forms of CKD, angiotension II plays the important role which its production and mechanistic target of rapamycin signaling maintains persistent podocyte hypertrophy and glomerular hyperfiltration and ultimately aggravates podocyte loss and proteinuria. Angiotensin II possibly also contributes to the dysregulated response of progenitor parietal epithelial cells along Bowman’s capsule, generating FSGS lesions instead of replacing lost podocytes [74]. This structural remodelling of the glomerulus presents clinically as proteinuria, which is a marker of nephron damage and is predictive of CKD progression [75]. on page 6, lines 216-236. 

Point 5: Figure 1: please explain with full names of all abbreviations in the figure legend. Additionally, several abbreviations are missing in the figure legend (e.g., PHD).

Response 5: Thanks for reviewer’s comments. We have added the full name of abbreviations in the figure legend on page 7, lines 253-258. 

Reviewer 3 Report

The review entitled "Kidney Injuries and Evolution of Chronic Kidney Diseases Due to Neonatal hyperoxia Exposure based on Animal Studies" by authors is well written with high level scientific content in this field.

However, introduction part need to be improve with more content so that its readability can increase. I recommend a picture or graphical representation of kidney injuries and evolution of Chronic kidney diseases would be easier to understand the trends but not necessary.  More keywords can be added. Short conclusion may be a good addition to this review.

I will recommend this review to publish in International Journal of Molecular Sciences after considering my minor comments.

Author Response

Response to Reviewer 3 Comments

The review entitled "Kidney Injuries and Evolution of Chronic Kidney Diseases Due to Neonatal hyperoxia Exposure based on Animal Studies" by authors is well written with high level scientific content in this field.

Point 1: However, introduction part need to be improve with more content so that its readability can increase. I recommend a picture or graphical representation of kidney injuries and evolution of Chronic kidney diseases would be easier to understand the trends but not necessary.  More keywords can be added. Short conclusion may be a good addition to this review.

Response 1:. Thanks for reviewer’s comments. We have added keywords on page 1, line 23, and short conclusion on page 10, lines 396-404.

Point 2: I will recommend this review to publish in International Journal of Molecular Sciences after considering my minor comments.

Response 2: Thanks for reviewer’s comments.

Round 2

Reviewer 2 Report

The authors answered all of my questions and revised the MS accordingly.

I suggest the acceptance of their MS for publication.